# A Cambrian origin for vertebrate rods

**Sabrina Asteriti[1], Sten Grillner[2], Lorenzo Cangiano[1]***

[1]Department of Translational Research, University of Pisa, Pisa, Italy; [2]Department of Neuroscience, Karolinska Institute, Stockholm, Sweden

**Abstract** Vertebrates acquired dim-light vision when an ancestral cone evolved into the rod photoreceptor at an unknown stage preceding the last common ancestor of extant jawed vertebrates (~420 million years ago Ma). The jawless lampreys provide a unique opportunity to constrain the timing of this advance, as their line diverged ~505 Ma and later displayed high-morphological stability. We recorded with patch electrodes the inner segment photovoltages and with suction electrodes the outer segment photocurrents of *Lampetra fluviatilis* retinal photoreceptors. Several key functional features of jawed vertebrate rods are present in their phylogenetically homologous photoreceptors in lamprey: crucially, the efficient amplification of the effect of single photons, measured by multiple parameters, and the flow of rod signals into cones. These results make convergent evolution in the jawless and jawed vertebrate lines unlikely and indicate an early origin of rods, implying strong selective pressure toward dim-light vision in Cambrian ecosystems.

## Introduction

The fossil record shows that by the middle Cambrian, camera-type eyes were already present in stem vertebrates (*Morris and Caron, 2014*), supporting the emerging concept that spatially resolved vision provided a major competitive advantage in those biota (*Paterson et al., 2011*). Lampreys, the only surviving jawless vertebrates together with the related hagfish (*Heimberg et al., 2010*), are a pivotal resource for gaining further insight into early vertebrate vision. In fact, their line diverged during the Cambrian (~505 Ma [*Erwin et al., 2011*]) and they later remained remarkably stable. This is true both of their external morphology, as revealed by fossil specimens (*Janvier and Arsenault, 2002*; *Gess et al., 2006*; *Janvier et al., 2006*; *Chang et al., 2014*), and of their anatomy, as demonstrated by primeval features such as the absence of bilateral limbs and of myelinated axons, and by their possession of the simplest nervous system among vertebrates. Adult lampreys have camera-type eyes with layered retinas containing all the major neuronal classes present in jawed vertebrates (*Lamb, 2013*) and sending retinotopically organized projections to the tectum (*Jones et al., 2009*), as well as a photosensory pineal organ (*Pu and Dowling, 1981*). Researchers have debated the rod or cone nature of lamprey retinal photoreceptors since the middle of the 19th century (relevant literature reviewed by *Walls, 1935*; *Govardovskii and Lychakov, 1984*; *Collin et al., 2009*) to ascertain whether, in vertebrates, cones pre-dated rods or vice versa. Current molecular genetic evidence indicates that modern rods evolved from an ancestral cone (*Okano et al., 1992*; *Yokoyama, 2000*; *Lamb et al., 2007*; *Kawamura and Tachibanaki, 2008*; *Shichida and Matsuyama, 2009*), implying that vision in near darkness is a relatively recent acquisition (*Lamb, 2013*) and causing the point of contention to become that of the timing of rod evolutionary emergence. This advance must have occurred (*i*) after the appearance of the precursor of rhodopsin and of other rod-specific phototransduction proteins isoforms and (*ii*) before the initial diversification of extant jawed vertebrates (~420 Ma; *Erwin et al., 2011*) endowed with modern rods. Phylogenetic analysis of visual opsins constrains time bound *i* to have occurred anywhere between the divergence of ascidians (~610 Ma; *Erwin et al., 2011*) and that of the lamprey line (~505 Ma; *Erwin et al., 2011*): the sea

*For correspondence: lorenzo. cangiano@unipi.it

**Competing interests:** The authors declare that no competing interests exist.

**eLife digest** The eyes of humans and many other animals with backbones contain two different types of cells that can detect light, which are known as rod and cone cells. Rod cells are much more sensitive to light than cone cells. The rods allow us to see in dim light by amplifying weak light signals and transmitting information to other cells, including the cones themselves. It is thought that the rod cell evolved from the cone cell in the common ancestors of mammals, fish, and other animals with backbones and jaws at least 420 million years ago.

Lampreys are jawless fish that diverged from the ancestors of jawed animals around 505 million years ago, in the middle of a period of great evolutionary innovation called the Cambrian. They have changed relatively little since that time so they provide a snapshot of what our ancestors' eyes might have been like back then. Like the rod and cone cells of jawed animals, the eyes of adult lampreys also have two types of photoreceptors. However, it was not clear whether the lamprey photoreceptor cells work in a similar way to rod and cone cells. Asteriti et al. collected lampreys in Sweden and France during their breeding season and used patch and suction electrodes to measure the activity of their photoreceptor cells. The experiments show that the short photoreceptor cells are more sensitive to light than the long photoreceptors and are able to amplify weak light signals. Also, the short photoreceptors send signals to the long photoreceptors in a similar way to how rod cells send information to cone cells.

The similarities between lamprey photoreceptor cells and those of jawed animals support the idea that they have a common origin in evolutionary history. Therefore, Asteriti et al. conclude that the ability to see in low light evolved before these groups of animals diverged about 505 million years ago. The picture that emerges is one in which our remote ancestors inhabiting the Cambrian seas already possessed dim-light vision. This would have allowed them to colonize deep waters or to move at twilight, an adaptation suggestive of intense competition or predation from other life forms.

squirt *Ciona intestinalis* has only one jawed vertebrate-related visual opsin (*Kusakabe et al., 2001*), while some lamprey species have all five major classes (*Yokoyama, 2000*) including an Rh1 rhodopsin ortholog (*Pisani et al., 2006*) (but see *Collin et al., 2003*). Recently, strong evidence has emerged indicating that these five opsin classes (and the rod-specific molecular toolbox) emerged in the context of two rounds of whole-genome duplication called '2R' (*Kuraku et al., 2009*; *Lagman et al., 2013*). Furthermore, analysis of the whole sea lamprey genome suggests that the lamprey line diverged from the main vertebrate line *shortly after* 2R (*Smith et al., 2013*). Therefore, unveiling the functional properties of lamprey photoreceptors may shed light on the evolution of dim-light vision in the critical time period following 2R (*Collin et al., 2009*; *Lamb, 2013*).

The two types of photoreceptors in the retina of Northern hemisphere lampreys have light-absorbing outer segments arranged in adjacent tiers (*Figure 1A*): those of short photoreceptors (SPs) lie in an inner tier, while those of long photoreceptors (LPs) lie in an outer tier, next to the pigment epithelium. This nomenclature is based on the entire length of the photoreceptors that of the outer segments showing instead the reverse pattern. Importantly, SPs express an Rh1 rhodopsin ortholog (*Pisani et al., 2006*) and some of their phototransduction protein isoforms examined thus far clade with those of rods (*Muradov et al., 2008*), but they also have molecular and morphological features of cones including outer segment discs that appear continuous with the plasma membrane (*Dickson and Graves, 1979*). Thus, while they retain archaic features of a cone progenitor, SPs are homologues of jawed vertebrate rods (*Lamb, 2013*). LPs, on the other hand, express an LWS red cone opsin and have a molecular fingerprint consistent with cones (*Muradov et al., 2008*). Here, we examined single lamprey photoreceptors at the levels of their inner and outer segments using two different recording techniques that provide complementary information, to establish the extent to which SPs operate like jawed vertebrate rods. We found multiple striking similarities that, taken together, argue against convergent evolution, implying that middle Cambrian vertebrates possessed functionally advanced rod precursors.

## Results

Using *Lampetra fluviatilis*, collected in Sweden and France during their spawning run, we investigated the function of photoreceptors in retinal slices maintained at a physiological temperature of 9–11°C.

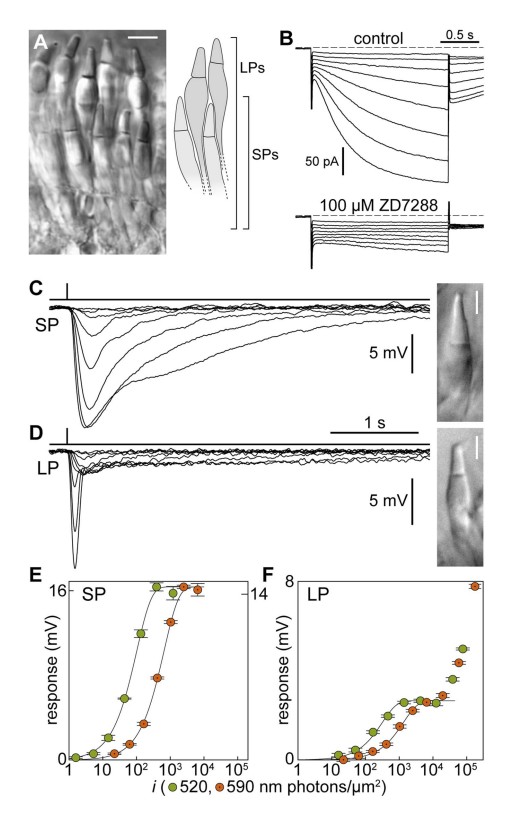

**Figure 1**. Signal processing in the inner segment of lamprey photoreceptors resembles that found in jawed vertebrates. (**A**) Image of a live retinal slice showing the layered organization of lamprey photoreceptors: short photoreceptors (SPs) in an inner tier and long photoreceptors (LPs) in an outer tier. Scale bar 10 μm. (**B**) Photoreceptors express the $I_h$ current: membrane current of a SP in response to hyperpolarizing voltage clamp steps (from a holding potential of −53 mV to −60/−67/−74/−81/−88/−95/−102/−109 mV and repolarization to −65 mV) in control and during superfusion of the $I_h$ blocker ZD7288 at 100 μM. Records are not averages. (**C–F**) Photovoltage responses reveal that SPs feed their signals into LPs. (**C** and **D**) Average responses to 520-nm flashes of a SP (0.5, 1.6, 5.4, 15, 45, 136, 398, 1128 photons·μm⁻²) and a LP (16, 51, 170, 469, 1413, 4314, 12,597, 38,847, 77,695 ph·μm⁻²). Insets show their outer segments (scale bars 5 μm). (**E** and **F**) Response amplitudes to 520-nm (green circles) and 590-nm flashes (orange circles with a dot) of a SP and a LP. Fits are exponential saturation functions (for the LP restricted to the first component: see text). In panel **E**, left and right ordinate values refer to left and right data sets, respectively, which were adjusted to match saturating amplitudes. In panel **F** such an adjustment could not be performed. Error bars are SEM. Action spectra templates for SPs and LPs are shown in *Figure 1—figure supplement 1*.

*Figure 1. continued on next page*

## Dark membrane potentials and inner segment properties of SPs and LPs

First, we made perforated patch-clamp recordings from photoreceptor inner segments and found that the dark membrane potential was of −43.2 ± 0.7 mV for SPs (n = 30) and −45.9 ± 1.1 mV for LPs (n = 10) (*Table 1*); these values are in line with those of jawed vertebrate rods and cones (*Cangiano et al., 2012*). Input resistances were 518 ± 41 MΩ (n = 8; SPs) and 442 ± 68 MΩ (n = 9; LPs). The membrane time constants, obtained by fitting single exponentials to the early rise of a current step response, were 31.9 ± 4.9 ms (n = 8; SPs) and 12.9 ± 1.3 ms (n = 9; LPs) (p < 0.001), equivalent to low-pass filtering with cut-off frequencies of ~5 Hz for SPs and ~12 Hz for LPs. Thus, the electrical properties of the inner segments of SPs seem adapted to process slower photocurrent changes than those of LPs. Both SPs (n = 7) and LPs (n = 2) expressed the hyperpolarization-activated current $I_h$, similarly to rods and cones (*Della Santina et al., 2012*); $I_h$ was abolished by ZD7288 (100 μM, n = 1 SP; *Figure 1B*).

## SPs feed their signals to LPs

Light stimulation evoked a hyperpolarization in both photoreceptors (*Figure 1C,D*), with peak changes in membrane potential of up to 30 mV (SPs) and 32 mV (LPs) in response to saturating flashes. The amplitudes of the flash responses from SPs were described by exponential saturation functions (*Figure 1C,E*). From the curves, we obtained a ratio of 4.4 ± 0.9 (n = 5) for the sensitivities of these photoreceptors at 520 nm and 590 nm. This value is in reasonably good agreement with the ratio of 5.6 predicted by an 11A₁ visual pigment template (*Govardovskii et al., 2000*) having a λ_max of 517 nm (*Figure 1—figure supplement 1A*), the absorbance maximum of SP outer segments found with microspectrophotometry (*Govardovskii and Lychakov, 1984*), and is thus consistent with the expression of an Rh1 visual pigment. For LPs, the flash responses displayed two components (*Figure 1D,F*): the first component had kinetics, sensitivity, and spectral preference similar to SPs; the second component had faster kinetics, lower sensitivity, and a ratio of sensitivities at 520 nm and 590 nm of 1.1 ± 0.04 (n = 6). This ratio agrees with the value of 1.1 predicted by an 11A₁ template (*Govardovskii et al., 2000*), whose λ_max is set at 555 nm (*Figure 1—figure supplement 1A*), the absorbance maximum of LP outer segments (*Govardovskii and Lychakov, 1984*), consistent

*Figure 1. Continued*

The following figure supplement is available for figure 1:

**Figure supplement 1**. Predicted action spectra of SPs and LPs and their relative sensitivities at 520 and 590 nm. DOI: 10.7554/eLife.07166.004

with their expression of an LWS pigment. It is likely that the first component of the flash response from LPs represents input from SPs, probably mediated by gap junctions; this arrangement would represent in lamprey retina an arrangement homologous to rod-cone coupling in jawed vertebrates (*Asteriti et al., 2014*). In support of this interpretation, we observed with Lucifer Yellow injection thin *telodendria* emanating from the synaptic pole of the photoreceptors and extending laterally into the inner plexiform layer (*Figure 2*): the only known function of these processes in jawed vertebrates is that of forming inter-photoreceptor junctional contacts (*O'Brien et al., 2012*). We attempted to uncouple these cells pharmacologically with MFA (100 µM) or 2-APB (10–20 µM), known blockers of retinal gap junctions, but unfortunately these agents produced marked non-specific effects (not shown).

## SPs display bleaching desensitization

During their spawning run, lampreys do not feed and rely exclusively on stored reserves for several months, leading to a vitamin A deficiency (*Wald, 1942*) that could hinder visual pigment regeneration.

**Table 1**. Electrophysiological parameters of SPs and LPs listed in the order they appear in the main text

| Parameter | Patch clamp | | Suction electrode |
| --- | --- | --- | --- |
| | SPs | LPs | SPs |
| $V_{dark}$ (mV) | $-43.2 \pm 0.7$ (n = 30) | $-45.9 \pm 1.1$ (n = 10) | – |
| $IR_{membrane}$ (MΩ) | $518 \pm 41$ (n = 8) | $442 \pm 68$ (n = 9) | – |
| $\tau_{membrane}$ (ms) | $31.9 \pm 4.9$ (n = 8)[]$_a$ | $12.9 \pm 1.3$ (n = 9) [***]$_a$ | – |
| Max response (mv) | 30 | 32 | – |
| $i_{1/2}$ (520 nm ph·µm$^{-2}$); *control* | $149 \pm 25$ (n = 10) []$_b$ | $1.9 \times 10^5 \pm 1.1 \times 10^5$ (n = 7) []$_c$ | – |
| $i_{1/2}$ (520 nm ph·µm$^{-2}$); *regenerated* | $63 \pm 11$ (n = 12) [**]$_b$ | $2385 \pm 513$ (n = 7) [*]$_c$ | – |
| Sensitivity 520/590; *control* | $4.4 \pm 0.9$ (n = 5) []$_d$ | $1.1 \pm 0.04$ (n = 6) []$_e$ | – |
| Sensitivity 520/590; *regenerated* | $5.6 \pm 1.0$ (n = 5) [n.s.]$_d$ | $1.8 \pm 0.1$ (n = 7) [**]$_e$ | – |
| Integration time, dim flash (s); *control* | $0.32 \pm 0.05$ (n = 13) []$_f$ | – | – |
| Integration time, dim flash (s); *regenerated* | $0.81 \pm 0.14$ (n = 9) [***]$_f$ | – | $1.45 \pm 0.10$ (n = 10) |
| TTP at $i_{1/2}$ (s); *regenerated* | $0.29 \pm 0.04$ (n = 11) []$_g$ | $0.11 \pm 0.007$ (n = 10) [***]$_g$ | – |
| $\tau_{rec}$ at $i_{1/2}$ (s); *regenerated* | $1.05 \pm 0.27$ (n = 11) []$_h$ | $0.12 \pm 0.02$ (n = 10) [***]$_h$ | – |
| $i_{1/2}$ ($\lambda_{max}$ ph·µm$^{-2}$); *regenerated* | $63 \pm 11$ (n = 12) []$_i$ | $777 \pm 167$ (n = 7) [***]$_i$ | – |
| Dim-flash sensitivity (mV·ph$^{-1}$·µm$^2$); *regenerated* | $0.61 \pm 0.17$ (n = 8) | – | – |
| Dim-flash sensitivity (%·ph$^{-1}$·µm$^2$); *regenerated* | $3.0 \pm 0.6$ (n = 8) | – | – |
| $a$ (pA); *regenerated* | – | – | $0.41 \pm 0.04$ (n = 10) |
| $a_{\%}$ (%·R$^{*\ -1}$); *regenerated* | $2.6 \pm 0.5$ (n = 8) | – | $2.6 \pm 0.3$ (n = 10) |
| SNR; *regenerated* | – | – | $1.5 \pm 0.1$ (n = 10) |
| $I_{dark}$ (pA); *regenerated* | $13 \pm 3$ (n = 4) | – | $16 \pm 1$ (n = 10) |
| Collecting area (µm$^2$·R*·ph$^{-1}$); *regenerated* | – | – | $0.83 \pm 0.17$ (n = 10) |
| Amplification constant (s$^{-2}$); *regenerated* | – | – | $0.59 \pm 0.09$ (n = 10) |

Values are given as 'mean ± SEM (sample size) [statistical significance]$_{identifier\ letter}$'; n.s.: not significant; *p < 0.05; **p < 0.01; ***p < 0.001; $V_{dark}$: dark membrane potential; $IR_{membrane}$: input resistance; $\tau_{membrane}$: membrane time constant; $i_{1/2}$: half-maximal response flash strength; TTP: time-to-peak; $\tau_{rec}$: decay time constant; *a*: absolute single photon response; $a_{\%}$: fractional single photon response; SNR: signal-to-noise ratio; $I_{dark}$: dark current; SPs: short photoreceptors; LPs: long photoreceptors.

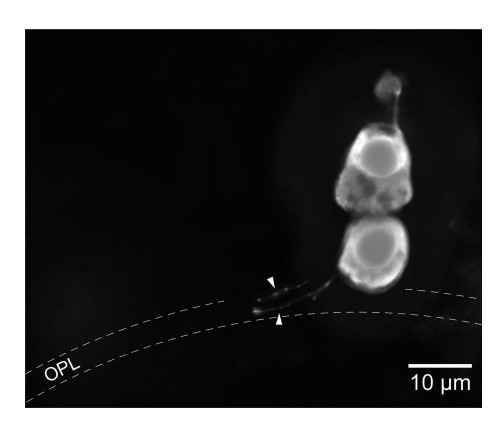

**Figure 2**. Lamprey photoreceptors extend telodendrial processes. An example of a lucifer yellow stain of a live SP showing two thin processes (arrowheads) extending laterally from the synaptic pole into the outer plexiform layer.

We thus wondered whether some of the visual pigment in our preparations might have been in a bleached state (i.e., devoid of its light-sensing chromophore). Clarifying this point was a crucial prerequisite to our subsequent assessment of single photon processing by SPs, for reasons explained in the rest of this paragraph. In jawed vertebrates, bleached rod and cone opsins constitutively activate the phototransductive cascade at a very low rate (*Cornwall and Fain, 1994*; *Cornwall et al., 1995*). Due to this property in rods, in which pigment regeneration is much slower than in cones, bleaches of even a small fraction of the total pigment pool caused by bright light lead to a significant and long-lasting desensitization, which is much larger than what is expected from the simple decrease in light-sensitive visual pigment molecules (*Fain et al., 2001*; *Lamb and Pugh, 2004*). Bleaching desensitization thus leads to a reduction in photo-transduction gain, and therefore, in the single photon response amplitude. Assuming that lamprey opsins behave similarly to those of jawed vertebrates, the possible presence of bleached visual pigment in our experiments (see above) raises the possibility that SPs were desensitized relative to their full potential.

To examine whether this was the case, we regenerated any bleached visual pigment molecules by superfusing the retinal slices in the recording chamber with the artificial analog 9-cis-Retinal (100 µM for 20–25 min). The sensitivity at 520 nm of two SPs, recorded both in control and during delivery of 9-cis-Retinal, increased by 2.0 and 2.5-fold (*Figure 3A*). Moreover, sensitivity was higher (p < 0.01) in regenerated than in control SPs: half-maximal response at 520 nm evoked with flashes ($i_{1/2}$) of $63 \pm 11$ photons·µm$^{-2}$ (n = 12; regenerated) vs $149 \pm 25$ photons·µm$^{-2}$ (n = 10; control). Thus, some of the visual pigment molecules in SP outer segments were indeed bleached. The ratio of sensitivities at 520

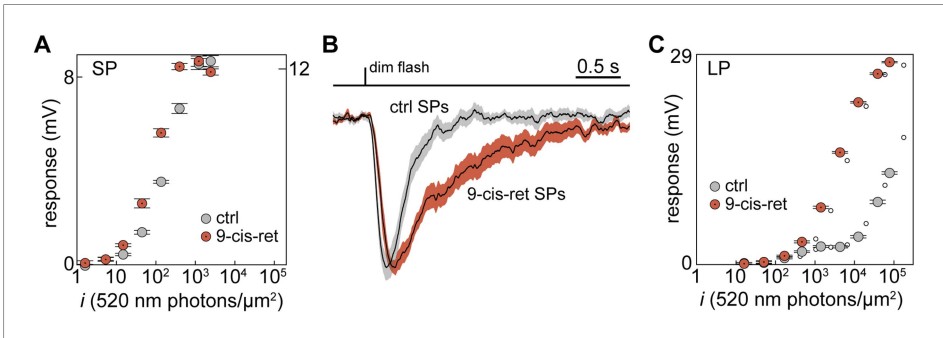

**Figure 3**. Visual pigment regeneration reveals the full sensitivity of photoreceptors in the upstream migrating river lamprey. (**A**) Photovoltage response amplitudes to 520-nm flashes before (gray circles) and after visual pigment regeneration with 9-cis-Retinal (red circles with a dot) of a SP. Left and right ordinate values refer to left and right data sets, respectively, which were adjusted to match saturating amplitudes. (**B**) Normalized-averaged-normalized dim-flash photovoltage responses in control (n = 13) and regenerated SPs (n = 9), highlighting the difference in integration time. These records were obtained as follows: (*i*) the average dim-flash response of each SP was normalized to its peak amplitude (always below 2 mV), (*ii*) normalized responses were averaged across cells, (*iii*) the final average was normalized to its peak. Shaded areas show ±1 SEM. (**C**) Photovoltage response amplitudes to 520-nm flashes before (gray circles) and after visual pigment regeneration with 9-cis-Retinal (red circles with a dot) of a LP. Responses to 590-nm flashes are also shown (small empty circles; error bars are smaller than circle diameter). Error bars are SEM.

and 590 nm did not differ significantly (p = 0.42) between regenerated and control SPs: $5.6 \pm 1.0$ (n = 5; regenerated) vs $4.4 \pm 0.9$ (n = 5; control). To test whether such bleaching was associated to desensitization, we examined dim-flash integration time (*Jones et al., 1996*). In one SP, integration time increased by 2.6-fold after superfusion with 9-cis-Retinal and the same parameter was significantly higher (p < 0.001) in regenerated than in control SPs (*Figure 3B*): $0.81 \pm 0.14$ s (n = 9; regenerated) vs $0.32 \pm 0.05$ s (n = 13; control). These results strongly suggest that SPs were in a state of bleaching desensitization.

Although secondary to the main goal of this analysis, we also tested the effect of 9-cis-Retinal on LPs. In one LP, the sensitivity at 520 nm increased by 22-fold after superfusion with 9-cis-Retinal (*Figure 3C*; second, lower sensitivity component). Moreover, sensitivity was much higher (p < 0.05) in regenerated than in control LPs: half-maximal response at 520 nm evoked with flashes ($i_{1/2}$) of $2385 \pm 513$ photons·µm$^{-2}$ (n = 7; regenerated) vs $1.9 \times 10^5 \pm 1.1 \times 10^5$ photons·µm$^{-2}$ (n = 7; control; second, lower sensitivity component; see 'Materials and methods' for details on the uncertainty of this specific value). As expected for the incorporation of 9-cis-Retinal in a significant fraction of the LP visual pigment pool (*Makino et al., 1999*), this increase in sensitivity was associated with a marked hypsochromic shift. Specifically, the ratio of sensitivities at 520 and 590 nm went from 1.1 to 1.7 in the single LP treated with 9-cis-Retinal (*Figure 3C*, compare large and small circles) and was significantly higher (p < 0.01) in regenerated than in control LPs: $1.8 \pm 0.1$ (n = 7; regenerated) vs $1.1 \pm 0.04$ (n = 6; control).

To examine the properties of lamprey photoreceptors under fully dark-adapted conditions, all subsequent experiments were made on retinas pretreated with 9-cis-Retinal.

## SPs are intrinsically slower than LPs

After pigment regeneration, the photovoltage responses of SPs, recorded with patch clamp, remained markedly slower than those of LPs (*Figure 4A,B*). To characterize the photoreceptors' kinetics, for each recorded cell we plotted time-to-peak (TTP) and decay time constant ($\tau_{rec}$) as a function of flash strength normalized to its half-maximal value ($i_{1/2}$) (*Figure 4C,D*). From linear fits to the data, we estimated the values of these parameters at $i_{1/2}$: for SPs, the TTP was $0.29 \pm 0.04$ s (n = 11) and the $\tau_{rec}$ was $1.05 \pm 0.27$ s (n = 11). In contrast, for LPs, the TTP was only $0.11 \pm 0.007$ s (n = 10; p < 0.001) and the $\tau_{rec}$ only $0.12 \pm 0.02$ s (n = 10; p < 0.001). Therefore, when compared at flash strengths eliciting responses of similar fractional amplitude, SPs were indeed slower than LPs. Importantly, while the estimates of TTP and $\tau_{rec}$ in LPs may have been influenced to some degree by the signals that are fed to them from SPs (see above), the latter would have acted to reduce (rather than increase) the differences in kinetics between the two photoreceptors.

## SPs are intrinsically more sensitive than LPs

To compare the intrinsic light sensitivity of regenerated SPs and LPs, we first considered whether we should correct their half-maximal flash strengths ($i_{1/2}$) measured at 520 nm for: (*i*) the position of the peaks of their action spectra ($\lambda_{max}$) with respect to the stimulus wavelength and (*ii*) the smaller quantum efficiency of pigment bound to 9-cis-Retinal (about one third; *Hubbard and Kropf, 1958*; *Hurley et al., 1977*). Both factors have the effect of reducing the sensitivity

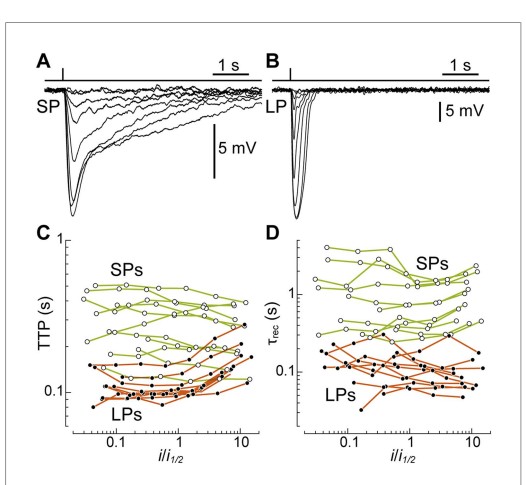

**Figure 4**. SPs are markedly slower than LPs. (**A** and **B**) Average responses to 520-nm flashes of a SP (0.5, 1.6, 5.4, 15, 45, 136, 398, 1128 photons·µm$^{-2}$) and a LP (16, 51, 170, 469, 1413, 4314, 12,597, 38,847, 77,695 ph·µm$^{-2}$), both recorded with patch clamp after visual pigment regeneration with 9-cis-Retinal. (**C** and **D**) Plots of time-to-peak (TTP) and decay time constant ($\tau_{rec}$) vs flash strength, normalized to its half-maximal value $i_{1/2}$, in regenerated SPs (n = 11; empty circles) and LPs (n = 10; full circles). The data from each cell are connected by lines.

displayed by the photoreceptor with respect to its maximum achievable level.

For SPs, we made the conservative assumption that they incorporated only a negligible amount of 9-cis-Retinal, as suggested by their limited increase in sensitivity following regeneration combined with their expression of bleaching desensitization (and supported by the non-significant change in their 520/590 nm sensitivity ratios). This implied that our 520-nm flashes essentially coincided with $\lambda_{max}$ (517 nm, see above) and that no correction was necessary for their $i_{1/2}$ of $63 \pm 11$ photons·$\mu m^{-2}$ (n = 12). Given the conservative nature of the above assumption, this value provides a *lower bound* for the maximal sensitivity of fully dark-adapted SPs.

For LPs, we made the equally conservative assumption that their entire visual pigment pool was replaced with 9-cis-Retinal. We then predicted their modified action spectrum (*Figure 1—figure supplement 1B*) by slightly adjusting two parameters of the $9A_1$ template for red cones of *Makino et al. (1999)* ($\lambda_{max\_A0}$ from 508 to 507 nm and $\lambda_{max\_G1}$ from 567 to 566 nm; see their Table 2) so as to match our experimentally determined ratio of sensitivities at 520 and 590 nm of $1.8 \pm 0.1$ (n = 7; see above). Taking into account the off-peak position of our flashes (520 nm) relative to the $\lambda_{max}$ of this action spectrum (541 nm) and the smaller quantum efficiency of regenerated pigment, the corrected $i_{1/2}$ of LPs was $777 \pm 167$ photons·$\mu m^{-2}$ (n = 7). Given the conservative initial assumption, this value provides an *upper bound* for the maximal sensitivity of fully dark-adapted LPs. The $i_{1/2}$ of LPs was much higher than that of SPs (p < 0.001). Note that signals feeding from SPs into LPs would have acted to reduce (rather than increase) the differences in sensitivity between the two photoreceptors, leaving our conclusions unchanged.

## The single photon response of SPs is within the range of jawed vertebrate rods

A crucial functional measure of the position of lamprey SPs with respect to the evolutionary transition from an ancestral cone to the modern rod is their performance in amplifying single photons (*Lamb, 2013*). Regenerated SPs were highly sensitive, with absolute and fractional dim-flash sensitivities in patch clamp of $0.61 \pm 0.17$ mV·photons$^{-1}$·$\mu m^2$ (n = 8) and $3.0 \pm 0.6\%$·photons$^{-1}$·$\mu m^2$ (n = 8). We obtained a first estimate of their fractional single photon response of $2.6 \pm 0.5\%$·$R^{*\ -1}$ (n = 8) by dividing the fractional dim-flash sensitivity with a theoretical effective collecting area of 1.18 $\mu m^2$·R*·photons$^{-1}$ (see 'Materials and methods'). A search for quantal responses using the patch-clamp technique proved inconclusive, as also observed in similar recordings of photo-receptors from those jawed vertebrates having extensively coupled rods (*Fain, 1975*). We thus performed suction electrode photocurrent recordings from the conical outer segments of SPs, in retinae pretreated with 9-cis-Retinal (100 $\mu M$ for 20–25 min): this recording technique only measures the current flowing through the membrane enclosed in the pipette and is thus ideally suited to examine phototransduction in a given photoreceptor without an appreciable contribution of its electrically coupled neighbors (*Baylor et al., 1979a*). Under these conditions, responses to repeated delivery of dim flashes were highly variable in amplitude (*Figure 5A*). We estimated the absolute amplitude of the single photon response (*a*) to be $0.41 \pm 0.04$ pA (n = 10), by dividing the increase in the time-dependent variance by the mean response for each SP (*Figure 5B*) (*Rieke and Baylor, 1998*) (for details on the use of variance analysis in single photon response estimation see the 'Materials and methods'). The normalized time-dependent squared mean responses and variance increases overlapped (*Figure 5B*) (*Rieke and Baylor, 1998*), consistent with the single photon responses being governed by Poisson statistics. The fractional amplitude of the single photon response (*a$_\%$*), determined for each SP on the basis of its maximal response to a single saturating flash delivered prior to the dim flash trains, was $2.6 \pm 0.3\%$·$R^{*\ -1}$ (n = 10), in line with the independent estimate obtained with patch (see above). Lastly, the signal-to-noise ratio (SNR), determined for each SP as the ratio of *a* over the standard deviation of the biological component of dark noise measured between consecutive dim flashes (0.5–20 Hz), was $1.5 \pm 0.1$ (n = 10). Importantly, the values of *a*, *a$_\%$*, and SNR in SPs are within the range reported for jawed vertebrate rods (*Figure 6*). In these experiments, we: (*i*) recorded only from intact outer segments (*Figure 1C*, inset), (*ii*) observed similar dark currents (maximum current change with a saturating flash) with patch clamp ($13 \pm 3$ pA, n = 4) and suction electrodes ($16 \pm 1$ pA, n = 10), and (*iii*) measured similar collecting areas ($0.83 \pm 0.17$ $\mu m^2$·R*·photons$^{-1}$, n = 10; ratio of the squared mean response over the product of the variance increase and the flash strength [*Rieke and Baylor, 1998*]) to theoretical prediction (1.18 $\mu m^2$·R*·photons$^{-1}$). From this, it is

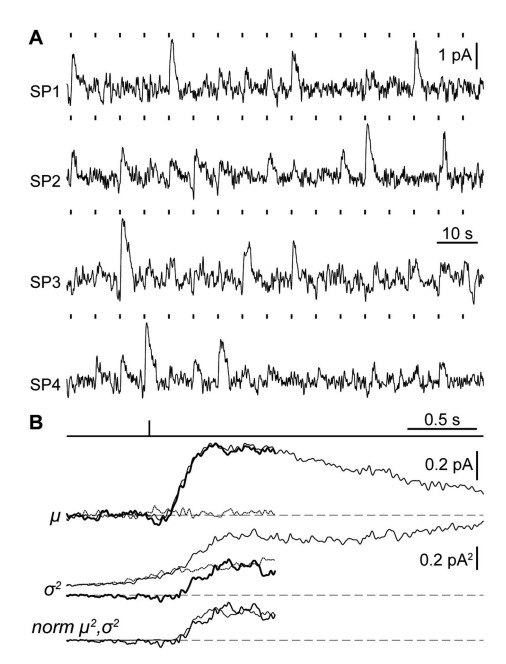

**Figure 5**. Suction electrode recordings of SP photocurrents in the single photon regime. (**A**) Samples of dim-flash response trains recorded from the outer segments of 4 SPs (SP1: flash strength = 1.64 photons·µm⁻², a = 0.25 pA; SP2: 3.26 ph·µm⁻², 0.43 pA; SP3: 1.64 ph·µm⁻², 0.56 pA; SP4: 3.26 ph µm⁻², 0.56·pA). (**B**) Single photon response analysis from one SP (SP3 in panel **A**), showing the mean response µ (thin trace: gross mean response, dotted trace: mean dark current, thick trace: net mean response), time-dependent variance σ² (thin trace: gross variance, dotted trace: dark current variance, thick trace: net variance), normalized squared mean response *norm* µ² (thin trace) and variance *norm* σ² (thick trace). Dashed lines indicate the current baseline or zero level. Dark current records were taken from the last 2 s preceding each flash and where baselined in the first 1.1 s, therefore, dark and net current records cover up to 0.9 s after the flash (see 'Materials and methods'). DOI: 10.7554/eLife.07166.008

highly likely that we had complete suction of the outer segment, and as such, we could make reliable estimates of $a_\%$ and SNR.

The single photon response analysis made with suction pipettes allowed us to estimate an amplification constant of phototransduction (*Pugh and Lamb, 1993*) in lamprey SPs of 0.59 ± 0.09 s⁻² at 9–11°C (n = 10), which lies between that of the large amphibian rods at room temperature (~0.1 s⁻²) and that of small mammalian rods at body temperature (~8 s⁻²) (*Lamb and Pugh, 2006*). The integration time measured with suction electrodes was 1.45 ± 0.10 s (n = 10), somewhat higher than that obtained with patch clamp (which also incorporates downstream processing in the inner segment). Taken together, the high-amplification constant and relatively long integration time are key contributors to SPs' single photon performance.

## Discussion

We examined lamprey SPs and LPs and found that their general properties closely match those of jawed vertebrate rods and cones: dark membrane potentials of around −45 mV, hyperpolarizing responses of up to 30 mV upon illumination, and the expression of a prominent hyperpolarization-activated $I_h$ current are all typical traits of jawed vertebrate photoreceptors and must therefore have been in place already by the middle Cambrian. A similar conclusion can be made for bleaching adaptation, which we confirmed to be present in SPs. In jawed vertebrates, this phenomenon is the result of a very low-constitutive activity of free opsin and occurs in both rods and cones exposed to bleaching lights (*Cornwall and Fain, 1994*; *Cornwall et al., 1995*; *Fain et al., 2001*; *Lamb and Pugh, 2004*). To assess whether SPs, the phylogenetic homologues of rods (*Pisani et al., 2006*; *Muradov et al., 2008*; *Lamb, 2013*), also operate like rods, we focused on multiple properties both at the level of single photon detection in the outer segment and of downstream signal processing in the inner segment. The rationale behind this approach is that, even if individual functional parameters of SPs and rods could have evolved independently towards a common present state (convergent evolution), this becomes quite unlikely if multiple common features are observed.

We find that SPs are exquisitely sensitive to light and feed their signals to the less sensitive but intrinsically faster LPs, similarly to the way rods feed their signals to cones via gap junctions in jawed vertebrates (*Asteriti et al., 2014*). The likely anatomical substrate of this signal crossover is represented by the thin telodendrial processes that we observed to extend laterally from the synaptic pole of the photoreceptors. Telodendria are ubiquitous in jawed vertebrates and their only known function is to form gap junctional contacts with nearby photoreceptors (*O'Brien et al., 2012*). While in mammals these processes are known to extend only from cone pedicles, in cold-blooded vertebrates they are also formed by rods (*Fadool, 2003*). Since rod-cone coupling is thought to provide a secondary route for rod signals when the high-gain output synapse of rods is saturated (*Attwell et al., 1987*), the question arises of whether lamprey SPs have properties of synaptic transmission

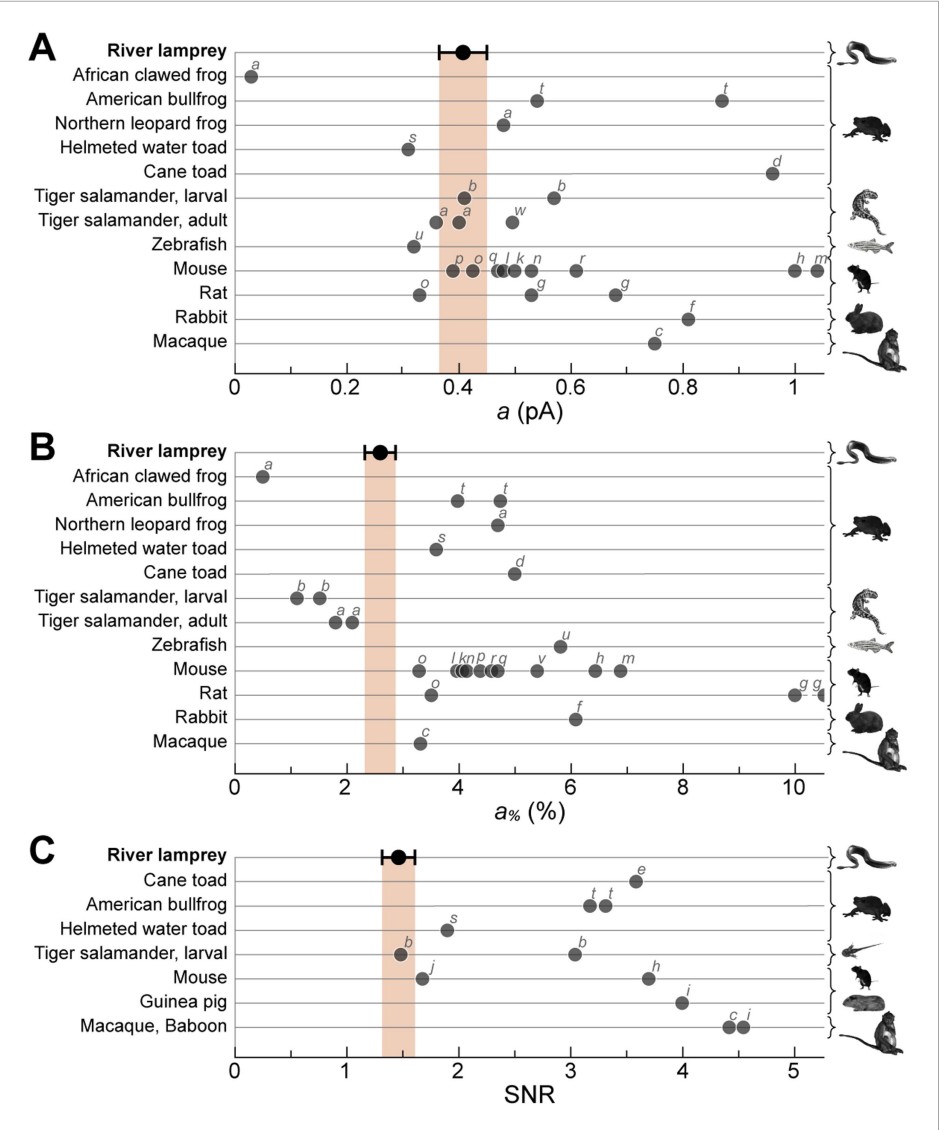

**Figure 6**. Lamprey SP single photon response parameters in the context of those of jawed vertebrate rods. River lamprey: this study; Jawed vertebrates: previous studies. (**A**) Absolute amplitude of the single photon response (*a*). (**B**) Fractional amplitude of the single photon response ($a_\%$). (**C**) Signal-to-noise ratio (SNR). Data points shown in the figure were obtained with suction pipette recordings of photoreceptor outer segments. Error bars for lamprey are SEM. Letters next to the data points correspond to the following references: a (*Palacios et al., 1998*), b (*Ala-Laurila et al., 2007*), c (*Baylor et al., 1984*), d (*Baylor et al., 1979a, 1979b*), e (*Baylor et al., 1980*), f (*Nakatani et al., 1991*), g (*Robinson et al., 1993*), h (*Field and Rieke, 2002b*), i (*Field and Rieke, 2002a*), j (*Okawa et al., 2010*), k (*Mendez et al., 2001*), l (*Burns et al., 2002*), m (*Azevedo and Rieke, 2011*), n (*Krispel et al., 2006*), o (*Luo and Yau, 2005*), p (*Makino et al., 2004*), q (*Wen et al., 2009*), r (*Gross and Burns, 2010*), s (*Palma et al., 2001*), t (*Donner et al., 1990*), u (*Vogalis et al., 2011*), v (*Nikonov et al., 2006*), w (*Rieke and Baylor, 2000*).

similar to those of rods and a dedicated postsynaptic circuitry for scotopic signaling. Furthermore, in mammalian photoreceptors, the $I_h$ current has been proposed to assist rod-cone signaling via gap junctions (*Seeliger et al., 2011*), a role it could also play in the lamprey retina given the presence of telodendria and SP–LP signal flow.

The functional similarities between lamprey SPs and jawed vertebrate rods discussed above, extend to the efficient processing in the single photon regime. With regards to scotopic performance, we found that dark-adapted SPs approach the efficiency of jawed vertebrate rods, both in terms of absolute and fractional single photon response and of its SNR. Here, it is important to note that while

the fractional single photon response $a_\%$ and the SNR are potentially subject to overestimation due to the fact that their denominators (the circulating current in darkness for $a_\%$ and the biological dark noise for the SNR) appear smaller if the outer segment is damaged or is not fully within the recording pipette (see however the evidence above that these conditions were respected in our experiments), this issue does not exist when estimating the absolute amplitude $a$.

Taken together, our findings raise the strong possibility that our last common ancestor in the middle Cambrian had already evolved scotopic vision. Recent evidence indicates that the lamprey line diverged *soon after* (*Smith et al., 2013*) the occurrence of two rounds of whole-genome duplication in stem vertebrates (2R) (*Dehal and Boore, 2005*; *Putnam et al., 2008*), which led to the diversification of the ancestral visual opsin and phototransduction protein repertoire (*Lagman et al., 2013*). Therefore, dim-light vision appears to have been acquired rapidly once a dedicated set of genes became available for specializing a type of cone into the rod (*Okano et al., 1992*; *Yokoyama, 2000*; *Lamb et al., 2007*; *Kawamura and Tachibanaki, 2008*; *Shichida and Matsuyama, 2009*). Admittedly, one cannot definitively exclude that such performance was refined independently in the two branches following their split ~505 Ma (*Erwin et al., 2011*): to do so, one would need to resurrect (*Thornton, 2004*) the full ancestral phototransductive cascade and then characterize its function. However, our data provide compelling support for the view that the evolution of the modern rod was already well under way at the time of divergence of the two branches: (*i*) the similarities between SPs and rods are multiple and extend from phototransduction to downstream processing, (*ii*) the photoreceptors specialized for scotopic vision in lampreys and in jawed vertebrates derive from one and the same precursor (and not from different ancestral cones), (*iii*) some of the phototransduction protein isoforms examined thus far in SPs clade with those of rods. This newly evolved ability to operate in dim light would have provided a significant advantage to early vertebrates faced with intense competition in a rapidly evolving ecological landscape (*Paterson et al., 2011*; *Lacalli, 2012*).

In coincidence with the submission of the present work, a study on the photoreceptors of a different species of lamprey was published (*Morshedian and Fain, 2015*), based on suction electrode recordings from outer segments. Although our study examines a broader range of photoreceptor properties, with both patch-clamp and suction electrode recordings, the two studies converge on the positive response to single photons.

## Materials and methods

*L. fluviatilis* of both sexes were collected during their spawning run either: (*i*) in the Swedish river Dal while migrating upstream from the gulf of Bothnia, at the end of summer 2013 or (*ii*) in the French river Garonne while migrating upstream from the Atlantic ocean, at the end of winter 2015. The animals were housed in a tank containing artificial water mimicking either: (*i*) that of the drainage lakes in the Dalarna region (*Arnemo, 1964*): (in µM) 60 $CaSO_4$, 110 $NaHCO_3$, 60 $MgCl_2$, 20 KCl, 290 $CaCO_3$ or (*ii*) that of the Garonne (*Etchanchu and Probst, 1988*): (in µM) 223 $CaSO_4$, 431 $NaHCO_3$, 200 $MgCl_2$, 42 KCl, 907 $CaCO_3$. Both water types were supplemented with 5% bottled mineral water for trace elements. Tank water temperature was kept at 5–6°C (i.e., below the spawning threshold of this species [*Hardisty and Potter, 1971*]) and steady state pH was 7.2. A 10-hr light/14-hr dark cycle was adopted, except in preparation for the electrophysiological experiment when the animals were dark-adapted for at least 24 hr. Lampreys were deeply anesthetized with 400 mg/l tricaine methanesulfonate (MS-222; E10521, Sigma–Aldrich, St. Louis, MO) and decapitated. All subsequent procedures were performed under dim far-red light in ice-cold bicarbonate buffered Ames' medium (A1420, Sigma–Aldrich), equilibrated with $O_2/CO_2$. Main constituents of Ames' are (in mM) 120 NaCl, 22.6 $NaHCO_3$, 6 D-glucose, 3.1 KCl, 1.2 $MgSO_4$, 1.1 $CaCl_2$, 0.5 $KH_2PO_4$, 0.5 L-glutamine. The head was pinned down in a sylgard-lined dissection chamber, the primary eye spectacles and the corneas were removed by performing circular cuts with fine scissors and the eyes enucleated with the lenses still in situ. One eyecup was transferred to the slicing chamber where the lens was removed and the retina gently detached from the sclera and extracted, while the other was stored in the fridge in bicarbonate buffered Ames' under an $O_2/CO_2$ atmosphere and used the following day. Slices of 250-µm thickness were obtained as previously described for the mouse retina (*Cangiano et al., 2012*), transferred to a recording chamber, superfused with bicarbonate buffered Ames' medium at 9–11°C, and visualized with DIC microscopy at 780 nm. For suction electrode recordings only, 0.25 mg ml$^{-1}$

hyaluronidase type IV (H3884; Sigma–Aldrich) was added to the slicing solution to clear the extracellular matrix surrounding the outer segments and facilitate pipette access.

## Perforated patch recordings

The inner segments of SPs and LPs lying close to the slice surface (*Figure 1A*) were visually targeted with 5–6 MOhm pipettes pulled with a P-97 (Sutter Instruments, Novato, CA) from borosilicate glass capillaries (1B120F-4, WPI, Sarasota, FL) and filled with a solution containing (in mM) 90 Kaspartate, 20 $K_2SO_4$, 15 KCl, 10 NaCl, 5 $K_2Pipes$, as well as 0.5 mg ml$^{-1}$ Lucifer yellow, and corrected to a pH of 7.20 with KOH/HCl. The backfilling solution also contained 0.4 mg ml$^{-1}$ Amphotericin B (item no. 11636, Cayman, Ann Arbor, MI) pre-dissolved in dimethyl sulfoxide (DMSO) at 60 mg ml$^{-1}$. Based on an analysis of the liquid junction and Donnan potentials when using this solution and recording photoreceptors (*Cangiano et al., 2012*), we report uncorrected values of membrane potential. Recordings were made with an Axopatch 1D amplifier, low-pass filtered at 500 Hz and acquired at 5 KHz with a Digidata 1320 and pClamp 9 software (Molecular Devices, Sunnyvale, CA).

## Suction electrode recordings

The outer segment current of intact SPs lying close to the slice surface (*Figure 1C*, inset) was recorded with suction electrodes (*Baylor et al., 1979a*). Specifically, glass capillaries (intraMARK, Blaubrand, Germany) were pulled with a P-97, broken to obtain even tips of 10–20 μm and heat polished to ~4-μm inner diameter. Tips were silanized by dipping in Sigmacote solution (SL2, Sigma–Aldrich), followed by vigorous back suction in air. Finally, pipettes were rinsed and filled with filtered Ames' medium, for a resistance in the bath of 2–3 MOhm. Intrapipette pressure was controlled with a pneumatic system filled with light mineral oil (330779; Sigma–Aldrich) using coarse and fine precision syringes (100 μL and 10 μL; Hamilton, Reno, NV) actuated by micrometer heads. Recordings were made in voltage clamp (holding voltage set at zero) with the same apparatus described above, except that signals were low-pass filtered at 20 Hz before acquisition (4-pole Bessel filter).

## Light stimulation

Full-field stimuli of unpolarized light were delivered by a green LED (520 nm; OD520; Optodiode Corp., Newbury Park, CA) or a yellow-orange LED (590 nm; APG2C3-590; Roithner LaserTechnik, Austria) mounted beside the objective turret. LEDs were driven by current sources commanded through the analog outputs of a Digidata 1320A (Axon Instruments, Foster City, CA). The power density reaching the recording chamber vs LED drive was measured separately with a calibrated low-power detector (1815-C/818-UV; Newport, Irvine, CA) positioned at the recording chamber. Flash duration was in the range 1–27 ms. Consecutive bright flashes were delivered at intervals of 13 s between each other. The photon flux density reaching the photoreceptors was derived from the measured power density and was likely to be overestimated to varying degrees across recorded cells due to reflection at the air–water interface and absorption by the surrounding tissue. In patch-clamp recordings, outer segments were generally, but not strictly, oriented at right angles with respect to the direction of incident light. In suction electrode recordings, orthogonality was instead guaranteed by the pipette itself.

## Pigment regeneration and pharmacology

Where specified, any bleached visual pigment was regenerated with the artificial chromophore analog 9-cis-Retinal. Stock solutions of 9-cis-Retinal (R5754; Sigma–Aldrich) in ethanol (100 mM) were prepared in darkness and stored at −80˚C. On the day of the experiment, an aliquot was thawed and diluted to a final concentration of 100 μM in Ames' medium integrated with 1% wt/vol fatty acid-free bovine serum albumin (A8806; Sigma–Aldrich), an effective solubilizing and protective agent (*Li et al., 1999*). This solution was delivered to the preparation directly in the recording chamber, without modifying flow rate or temperature, for 20–25 min followed by washout. A pharmacological blockade of gap junctions was attempted with meclofenamic acid (MFA; M4531, Sigma–Aldrich) and 2-aminoethyl diphenylborinate (2-APB; D9754, Sigma–Aldrich). The $I_h$ current was blocked with 4-Ethylphenylamino-1,2-dimethyl-6-methylaminopyrimidinium chloride (ZD7288; cat. no. 1000, Tocris, United Kingdom).

## Collecting area

The effective collecting area of SPs was estimated according to the approximate relation (*Baylor et al., 1979b*):

$$A_c = 2.303 \cdot V \cdot f \cdot \alpha \cdot Q_{isom}, \tag{1}$$

where V is the volume of their truncated conical outer segments, estimated at 102 μm$^3$ from mean values of length (13.0 μm), basal diameter (4.5 μm), and apical diameter (1.6 μm) measured in optical images acquired from live slices; *f* is a factor accounting for the dichroism of native opsin (1 for light incident along the outer segment axis and 0.5 for unpolarized light incident at right angles); α is the specific axial pigment density at the maximum absorption wavelength, measured (*Govardovskii and Lychakov, 1984*) in lamprey SPs at 0.015 μm$^{-1}$, a value similar to that of lower vertebrate rods containing rhodopsin and/or porphyropsin (*Harosi, 1975*); $Q_{isom}$ is the quantum efficiency of photoisomerization, assumed to have the rhodopsin value of 0.67 R*·photons$^{-1}$. Note that any bleached visual pigment regenerated with 9-cis-Retinal would have the somewhat different absorption spectrum and quantum efficiency of isorhodopsin (*Hubbard and Kropf, 1958*; *Hurley et al., 1977*; *Makino et al., 1999*); however, this contribution was neglected since the twofold–threefold increase in sensitivity displayed by SPs after delivery of 9-cis-Retinal could result from the regeneration of a small fraction of bleached pigment due to the phenomenon of *bleaching desensitization* (e.g., in larval tiger salamander rods, bleaching of ~6% rhodopsin halves sensitivity [*Jones et al., 1996*]).

## Data analysis and reporting

Data in the text and graphs are reported as mean and standard error of the mean. Statistical significance was assessed with the Mann–Whitney–Wilcoxon test. Integration time was defined as the integral of the dim-flash response divided by its peak amplitude. TTP was measured starting from the middle of the flash. Decay time constant ($\tau_{rec}$) was estimated by fitting a single exponential to the second half of the flash response recovery phase (i.e., starting from the point when the response had recovered to about half of its peak amplitude).

### Patch clamp

Half-maximal response flash strength ($i_{1/2}$) and light sensitivity (defined as its reciprocal) were obtained by fitting exponential saturation functions to response amplitude vs flash strength graphs. In the fitting procedure, the brightest saturating flashes delivered to SPs were excluded since they showed diminishing response amplitudes due to incomplete recovery from the previous flash in the 13-s interval used in our protocols. Since the low-sensitivity response component of LPs untreated with 9-cis-Retinal did not reach saturation with the flash strengths used in this study, fits were performed (*i*) by using the saturating amplitude observed after delivery of 9-cis-Retinal (when this was available) or (*ii*) by assuming a conservative saturating amplitude of 16 mV (i.e., the smallest value observed in LPs). The second assumption implies that we likely underestimated the half-maximal flash strength of unregenerated LPs (low-sensitivity component). Precise estimates of SP dim-flash sensitivity were obtained by dividing the amplitude of the mean response to many dim flashes by the strength of these flashes.

### Suction electrodes

The single photon response amplitude *a* was estimated by the established method of variance analysis (*Baylor et al., 1979b*, *1984*; *Rieke and Baylor, 1998*; *Vogalis et al., 2011*). Specifically, for each SP, the *net* mean response and time-dependent variance were calculated, from an ensemble of at least 50-dim flashes, as follows: (*i*) the *gross* mean response and variance were calculated after aligning the raw flash response records by subtracting their average value in the 1.1-s interval before the flash; (*ii*) the mean and variance in darkness were calculated from the 2-s interval prior to each flash, again after aligning the raw dark records by subtracting their average value in the first 1.1 s; (*iii*) the *net* mean response and variance were obtained by subtracting those in darkness from the *gross* ones, and therefore, include only the first 0.9 s after the flash: this, however, was sufficient to encompass the peak of the dim-flash response (*Figure 5B*). In those SPs in which dim flashes of two strengths were delivered, the final estimate of *a* was the average of the two separate estimates, weighed by their respective number of flashes. As in other rod studies in which *a* was small relative to total recorded dark noise, we could not perform a reliable analysis of the single photon response with amplitude histograms (notably, only 6 of the 26 estimates of *a* plotted in *Figure 6A* included histogram analysis). However, when both the

variance and histogram techniques were employed, similar values of $a$ were found (*Baylor et al., 1979b*, *1984*; *Vogalis et al., 2011*). Biological dark noise was taken as the square root of the difference between the recorded dark noise variance (integral within 0.5–20 Hz of the average power spectrum in the 2.1 s before the dim flashes) and the expected thermal noise variance (*Baylor et al., 1979b*, *1984*) (using the measured pipette seal resistance). The amplification constant of phototransduction was estimated by fitting the rising phase of the average dim-flash response with Equation 23 in *Pugh and Lamb (1993)*. For these fits, a delay $t_{eff}$ of 23 ms was used, which was separately estimated by fitting Equation 20 in *Pugh and Lamb (1993)* to full families of flash responses. The average number of photoisomerizations/flash was a result of the single photon response analysis.

## Data presentation

Exclusively for presentation purposes, the electrophysiological records shown in the figures were conditioned as follows: (*i*) current-clamp patch recordings were 'box car' filtered with a running window of 20 ms, (*ii*) voltage-clamp suction pipette recordings were detrended (*Vogalis et al., 2011*) to remove slow fluctuations in the baseline current and digitally low-pass filtered at 2 Hz.

## Acknowledgements

We thank Prof. Luigi Cervetto, Profs. Ellis and Linda Cooper, for providing comments on the manuscript, Dr Brita Robertson and Prof. Peter Wallén for assistance with shipping the Swedish lampreys, Prof. Ansgar Büschges and Dr Christoph Guschlbauer for generously providing us the French lampreys, Dr Monica Mazzolini for suggestions regarding the suction electrode recordings, Prof. Mario Pellegrino for access to his lab's microforge, and Mr Francesco Montanari for technical assistance.

## Additional information

### Funding

| Funder | Grant reference | Author |
| --- | --- | --- |
| Università di Pisa | Faculty of Medicine Bando Ricerca 2012/109 | Lorenzo Cangiano |

The funder had no role in study design, data collection and interpretation, or the decision to submit the work for publication.

### Author contributions

SA, Conception and design, Acquisition of data, Analysis and interpretation of data, Drafting or revising the article; SG, Analysis and interpretation of data, Contributed unpublished essential data or reagents; LC, Conception and design, Analysis and interpretation of data, Drafting or revising the article

### Ethics

Animal experimentation: All procedures involving the handling of experimental animals were approved by the Ethical Committee of the University of Pisa (prot. n. 2891/12) and were conducted in accordance with Italian (D.lgs.vo 116/92) and EU regulations (Council Directive 86/609/EEC).

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
