## [Decision Letter]

[Editors’ note: this article was originally rejected after discussions between the reviewers, but the authors were invited to resubmit after an appeal against the decision.]

Thank you for choosing to send your work entitled “A Cambrian Origin for Vertebrate Rods” for consideration at *eLife*. Your full submission has been evaluated by a Senior editor, a Reviewing editor, and three peer reviewers, and the decision was reached after discussions between the reviewers. Based on our discussions and the individual reviews below, we regret to inform you that your work will not be considered further for publication in *eLife*.

Three experts carefully reviewed the manuscript. While the reviewers found the work interesting, the number of substantive questions raised was such that we feel we must reject it. We hope that the reviewers’ comments below will be useful to you in revising the manuscript.

*Reviewer #1*:

The manuscript by Asteriti and colleagues describes experiments seeking to characterize the functional properties of lamprey SP and LP photoreceptors in an effort to determine the time of the evolutionary emergence of rods. They perform patch clamp and suction electrode recordings from SP and LP photoreceptors and show that these cells differ in their sensitivity and response kinetics, similarly to modern rods and cones. Notably, SP photoreceptors display fractional sensitivity and single photon response amplitude comparable to these of jawed vertebrate rods. The authors conclude that the origin of rods precedes the divergence of the lamprey line during the Cambrian.

This is an interesting result addressing a fundamental question in photoreceptor biology. Although a similar study was recently published by Morshedian and Fain in Current Biology (Curr Biol 25, 484-487, Feb 2015), this current study appears to be independent and also provides several original points. These include the coupling of SP and LP photoreceptors, which is similar to the one found in jawed vertebrates, and the functional effects of the seasonal lamprey vitamin A deprivation of SP and LP flash responses.

Although the data generally is consistent with the conclusions, the results are not presented in a systematic way and gaps in the analysis and presentation, together with the low numbers for some of the experiments raise doubts about the overall quality of the paper:

The authors mention that the averaged sensitivity is higher in 9-cis regenerated SPs compared to controls and even give a *p*-value and “*n*” but do not provide the actual sensitivity values. The same issue applies to the integration time. For LPs, the corresponding statement is restricted to one cell. Such glaring omission raises doubts about the quality of the data. The authors should address this by providing values for all measured parameters in SP and LP cells, ideally in a table format.

There is no mention whether the authors took into account the slow (presumably SP-driven) component of the flash response, as shown in Figure 2, when analyzing the LPs response sensitivity and kinetics.

The mean response and variance traces in Figure 5 are truncated shortly after the peak. Why?

“Duration” of the response is not typically used in photoreceptor response analysis. More rigorous ways of analyzing the kinetics of flash responses exist. In addition to integration time, mentioned by the authors, these include time to peak and recovery time (exponential fit to the tail of the response shutoff).

Amplification constant is mentioned in the Methods but not mentioned in the Results. What were the values for SPs and LPs and how did they compare to these of jawed vertebrate cells?

In the Discussion, bleaching adaptation is mentioned as one of the similarities between lamprey SPs and jawed rods. This is not a rod-specific phenomenon as cones (and apparently LPs) experience similar bleaching adaptation (e.g. Cornwall et al., GL. J. Gen Physiol. 1995. 106:543-57). In addition, the statement that bleached opsin is key determinant of the very slow time course of dark adaptation is misleading. Dark adaptation is regulated by the regeneration of visual pigment and the supply of chromophore (e.g. Lamb TD, Pugh EN Jr. Prog Retin Eye Res. 2004 23:307-80), which occur significantly slower for rods than for cones.

*Reviewer #2*:

This paper describes experiments on the light-response properties of retinal photoreceptors in the lamprey. The results and conclusions are similar to those published most recently by Morshedian and Fain in Current Biology (2015). Both groups studied the northern-hemisphere lamprey, albeit different species. Considering the close proximity in time between the two, the respective experiments probably proceeded pretty much in parallel.

The experiments are mostly fairly straightforward and standard, describing membrane parameters, membrane potential and such, as well as the light-response properties. The synaptic interactions between rod-like (SP) and cone-like (LP) photoreceptors (input from SP to LP cells) described here were apparently not observed by Morshedian and Fain. The reason for this is not entirely clear, but possibly due to different thicknesses of the retinal slices used by the two groups (thickness not specified by Morshedian and Fain) that might have disrupted such interactions in one but not the other preparation.

The section on bleaching adaptation is cryptic and not clear. For SP cells, it was described that the sensitivity was higher after regeneration by exogenous chromophore than in control. Considering that the retina was dark-adapted for 24 hours, and that rhodopsin (unlike cone pigment) does not show the property of re-dissociation (thus no chromophore exchange) in darkness, the increase in sensitivity after regeneration must reflect the presence of naked opsin already in darkness due to vitamin-A deficiency as the authors quoted, but they should at least make some clarifying comments and possibly even conduct more experiments to investigate further. Otherwise, the authors would come across as trivializing the phenomenon of bleaching adaptation (whereas it is in fact far from trivial). For LP cells, having done just one experiment is also unacceptable. In any case, what is the bottom line of this section in relation to the overall paper? This reviewer thinks that it should either be deleted completely or more experiments are necessary.

In the single-photon-response section, no response-amplitude histogram is shown, despite the many flash trails performed. This should be properly shown and compared with the estimate based on the variance/mean ratio.

Overall, some of the experiments appear to be hastily performed/analyzed and the paper superficially written, perhaps under pressure due to competition from the other group. Nonetheless, even if so, this is not acceptable for *eLife*, a journal that this reviewer understands to be of very high quality.

*Reviewer #3*:

In this paper, Asteriti et al. study the physiological characteristics of the photoreceptors of the northern hemisphere lamprey, *Lamptra fluviatilis*, and discuss when the “modern rods” evolved from an “ancestral cone.”

They have shown that 1) the short (SP) and long (LP) photoreceptors exhibit dark membrane potentials and inner segment properties of jawed vertebrate rods and cones, respectively, 2) visual pigments in SPs are maximally sensitive to 520 nm and those in LPs are sensitive to 590 nm, which were suspected to represent RH1 and LWS pigments, respectively, 3) 9-cis-retinal delivery to vitamin A1-starved SPs increased the efficiency in the sensitivity and time of presumed RH1 pigment response, 4) LPs respond to light at 520 nm much faster than SPs and 5) single photon responses of SPs correspond to those of jawed vertebrate rods. These results show that the lamprey uses photoreceptors with the characteristics of jawed vertebrate rods and cones and imply that “our last common ancestor in the middle Cambrian had already evolved scotopic vision” (Discussion, last paragraph).

These results improve our understanding of lamprey vision and the display of the cone-like morphologies of the SPs and LPs is effective (Figure 2). However, when the authors relate their results to photoreceptor evolution, the clarity of the presentation decreases. The problem is that rods and cones cannot be defined as clearly as the authors indicate. Ancestral photoreceptor cell types may be guessed from those of present-day lampreys, hagfishes, sea-squirts and lancelets, but we really don't know what they actually looked like. Photoreceptor cells in these modern organisms are the evolutionary products of the geological time scale and can differ significantly from those of their ancestors. The oft-discussed *Anolis carolinensis* with supposedly pure-cone retinas and *Gekko gecko* with presumably pure-rod retinas express “typical” rod- and cone-specific pigments, respectively. We can easily imagine photoreceptors with the mixture of rod- and cone-characteristics.

To consider photoreceptor evolution, the authors may relate their observations to visual pigment evolution because the functional changes of visual pigments are basically closely intertwined with photoreceptor evolution. For that, the following three observations may be of interest:

First, the extensive opsin sequence data show that visual pigments in vertebrates, including the lamprey, are classified into five evolutionarily distantly related RH1, RH2, SWS1, SWS2 and M/LWS pigment groups (e.g. Figure 22 in [47]). Such phylogenetic trees also show that all five groups of visual pigments already existed well before the lamprey ancestor appeared and, therefore, the lamprey may not be claimed to be the critical species to study photoreceptor evolution.

Second, by actually engineering the visual pigments at several stages of vertebrate evolution, it has been found that RH1 pigments (Yokoyama et al. PNAS 105, 13480-13485, 2008), SWS1 pigments (Shi & Yokoyama PNAS 100, 8308-8313, 2003) and M/LWS pigments (Yokoyama & Radlwimmer Genetics 158, 1697-1710, 2001) of most early vertebrate ancestors detected light maximally at 500 nm, 360 nm and 560 nm, respectively. These well-differentiated absorption spectra of visual pigments in the vertebrate ancestor further suggest that basic chemical machineries involved in the phototransduction also differentiated before the lamprey appeared.

Hence, the coexistence of rods and cones in the vertebrate ancestor, as implicated by the authors, is consistent with the functional evolutionary analyses of visual pigments. However, the reconstruction of ancestral visual pigments reveals nothing about photoreceptor cell types and the analyses of the SPs and LPs of the present-day northern hemisphere lamprey are meaningful.

Third, the five major visual pigment groups in vertebrates have arisen from a single type of pigment, followed by three rounds of gene duplication. At present, several visual pigments of sea-squirt and lancelet have been characterized, among which only one pigment in sea-squirt clusters with the vertebrate visual pigments (e.g. Kusakabe et al. FEBS Lett. 506, 69-72, 2001). Hence, if we are interested in the evolution of rods and cones, the critical time period may be somewhere after the appearance of lancelet but before the appearance of the lamprey, which is much earlier than the authors indicate. Then, the critical question seems to be what the photoreceptor that expressed this specific ancestral visual pigment looked like in the first place.

In summary, the physiological data of the SPs and LPs of the northern hemisphere lamprey obtained by the authors are valuable. However, when these results are interpreted as a part of photoreceptor evolution, the presentation becomes murky. For this reason, I cannot recommend that the paper be published as it stands.

---

## [Author Response]

We sincerely appreciate the time and effort that was devoted to this evaluation, as well as the fast turnaround time. We present our objections to the concerns raised by reviewers 2 and 3 below.

Reviewer 1, on the other hand, was supportive of the importance of our study and only found gaps in the analysis and presentation of the data, and a low number of some of the experiments. Here we also address his/her concerns, showing that the ‘missing’ analysis/data is either already present in the current manuscript or can be easily included.

Lastly, following submission of our manuscript to *eLife*, we became aware that a study on the photoreceptors of another species of lamprey had just appeared in Current Biology (Feb 16 issue; [57]; “Single-photon sensitivity of lamprey rods with cone-like outer segments”). Our study goes well beyond what is shown by these authors in many important respects (as also acknowledged by reviewer 1). By examining multiple aspects of lamprey photoreceptor function we provide a much stronger case for an early origin of functionally modern rods: while one individual property could have emerged independently in the jawless and jawed vertebrate lines through convergent evolution, the more properties one finds to match the less likely this becomes.

Reviewer #1:

*Although the data generally is consistent with the conclusions, the results are not presented in a systematic way and gaps in the analysis and presentation, together with the low numbers for some of the experiments raise doubts about the overall quality of the paper*:

*The authors mention that the averaged sensitivity is higher in 9-cis regenerated SPs compared to controls and even give a* p*-value and “*n*” but do not provide the actual sensitivity values. The same issue applies to the integration time*.

In the original manuscript (in the Results and Discussion sections) we did provide the values of both parameters for *regenerated* SPs: light sensitivity (“SPs had a half maximal response at 520 nm evoked with flashes of 43 ± 12 photons·μm^–2^ (n=6))” and integration time (“… a relatively long integration time of 1.45 ± 0.10 s (n=10))”. The values in control (vs. regenerated) SPs were omitted for brevity.

We have now included the absolute values of light sensitivity and integration time for control SPs (in the second paragraph of the subsection headed “SPs display bleaching desensitization”). Moreover, as a byproduct of additional experiments we have increased the sample size in both the control and regenerated groups. This data is also reported in a new Table 1.

*For LPs, the corresponding statement is restricted to one cell. Such glaring omission raises doubts about the quality of the data. The authors should address this by providing values for all measured parameters in SP and LP cells, ideally in a table format*.

The single value mentioned by the reviewer is the increase in sensitivity observed in one LP upon delivery of 9-cis-retinal. Not reported in the original manuscript was data from separate groups of control (n=4) and regenerated LPs (n=6), providing statistical support (p<0.01) for an effect of 9-cis-retinal on this subset of photoreceptors. The reason for such omission was that our main focus in this section was to establish whether SPs (the rod-like photoreceptors) in migrating lampreys are partly bleached due to a vitamin A deficiency, as originally suggested by [79]: obtaining this piece of information was essential to ensure that our subsequent single photon response analysis was performed in maximally sensitive (and not desensitized) SPs. Accordingly, the title of this section of the manuscript was “Bleaching adaptation (i.e. dark adaptation) in SPs”.

We have increased our sample size by performing new experiments and reported all available data on the effect of pigment regeneration in LPs (please see the subsection “SPs display bleaching desensitization”):

a) Sensitivity of regenerated (n=7) versus control LPs (n=7)

b) Sensitivity ratio 520/590 nm of regenerated (n=7) versus control LPs (n=6)

This data is also reported in a new Table 1.

*There is no mention whether the authors took into account the slow (presumably SP-driven) component of the flash response, as shown in*
Figure 2*, when analyzing the LPs response sensitivity and kinetics*.

As shown in a new Figure 4, in regenerated LPs the slow component cannot be distinguished from the fast one. Therefore, the values of sensitivity and kinetics that we report for LPs necessarily include some SP-mediated input. This, however, does not affect our conclusion that LPs are intrinsically faster and less sensitive than SPs: any SP-mediated input would make LPs appear more SPs-like and thereby act to reduce the difference between the two photoreceptors.

We have created two entirely new sections devoted to the analysis of the intrinsic differences in kinetics (“SPs are intrinsically slower than LPs”) and sensitivity (“SPs are intrinsically more sensitive than LPs”) between SPs and LPs. In both sections we explicitly mention the possible contribution of SP-LP signaling and present the argument outlined above.

*The mean response and variance traces in*
Figure 5
*are truncated shortly after the peak. Why*?

As mentioned in the original Methods section mean and variance of the raw dark records were obtained in the 2 s interval prior to each flash (aligned by subtracting their average value in the first 1.1 s). Since these were subtracted from the mean and variance of the flash responses, the final net records include only the first 0.9 s after the flash. As shown in Figure 5 this interval was sufficient to capture the peak of the dim flash response.

We have revised the manuscript to clarify this point, both in the legend of Figure 5 and in the Materials and methods (end of the subsection entitled “Data analysis and reporting”).

*“Duration” of the response is not typically used in photoreceptor response analysis. More rigorous ways of analyzing the kinetics of flash responses exist. In addition to integration time, mentioned by the authors, these include time to peak and recovery time (exponential fit to the tail of the response shutoff)*.

We agree with the reviewer that the duration of the flash response is seldom used in the literature, although it was rigorously defined in our manuscript.

As recommended by the reviewer, we have replaced this parameter with time-to-peak (TTP) and time constant of recovery (τ_rec_). The relevant data is now presented in a new dedicated section entitled “SPs are intrinsically slower than LPs”, as well as in a newly added Figure 4 (see figure file and legend).

*Amplification constant is mentioned in the Methods but not mentioned in the Results. What were the values for SPs and LPs and how did they compare to these of jawed vertebrate cells*?

In the original manuscript, we did report the amplification constant of SPs in the Results section: “The high amplification constant of phototransduction (69) of 0.59 ± 0.09 s^–2^ at 9–11°C (n=10)…”.

Providing an amplification constant for LPs is beyond the scope of our study, which is primarily concerned with characterizing the functional properties of SPs and comparing them to those of jawed vertebrate rods. Technically, it would be challenging to perform a fair comparison between SPs and LPs: while our estimate of the amplification constant in SPs is based on their measured collecting area (obtained from the variance analysis of the single photon response), an estimate in LPs would have to rely on a theoretically calculated collecting area.

In the revised manuscript we compare the amplification constant of SPs to other vertebrates (subsection headed “The single photon response of SPs is within the range of jawed vertebrate rods”), showing that it falls between that of large amphibian rods at room temperature (∼0.1 s^–2^) and small mammalian rods at body temperature (∼8 s^–2^). This parameter is strongly dependent on temperature and outer segment volume, and, for any given rod, it acquires significance in the context of single photon processing when viewed together with integration time (50). Accordingly, we report amplification constant and integration time together.

*In the Discussion, bleaching adaptation is mentioned as one of the similarities between lamprey SPs and jawed rods. This is not a rod-specific phenomenon as cones (and apparently LPs) experience similar bleaching adaptation (e.g. Cornwall et al., GL. J. Gen Physiol. 1995. 106:543-57)*.

*In addition, the statement that bleached opsin is key determinant of the very slow time course of dark adaptation is misleading. Dark adaptation is regulated by the regeneration of visual pigment and the supply of chromophore (e.g. Lamb TD, Pugh EN Jr. Prog Retin Eye Res. 2004 23:307-80), which occur significantly slower for rods than for cones*.

We agree with the reviewer that, while bleaching adaptation is prominent in jawed vertebrate rods, it is not exclusive to rods.

We have revised our text to convey the correct meaning that bleaching adaptation is present in SPs as it is in higher vertebrate rods and cones, but that recovery following bleach is slower in rods due slower pigment regeneration (Results, subsection headed “The single photon response of SPs is within the range of jawed vertebrate rods”, and Discussion, first paragraph).

Reviewer #2:

*This paper describes experiments on the light-response properties of retinal photoreceptors in the lamprey. The results and conclusions are similar to those published most recently by Morshedian and Fain in Current Biology (2015). Both groups studied the northern-hemisphere lamprey, albeit different species. Considering the close proximity in time between the two, the respective experiments probably proceeded pretty much in parallel*.

*The experiments are mostly fairly straightforward and standard, describing membrane parameters, membrane potential and such, as well as the light-response properties. The synaptic interactions between rod-like (SP) and cone-like (LP) photoreceptors (input from SP to LP cells) described here were apparently not observed by Morshedian and Fain. The reason for this is not entirely clear, but possibly due to different thicknesses of the retinal slices used by the two groups (thickness not specified by Morshedian and Fain) that might have disrupted such interactions in one but not the other preparation*.

There is a fundamental reason that explains why SP-LP interactions are not described in the Current Biology paper: their analysis is limited to phototransduction in the outer segment using suction electrode recordings, while we also investigated downstream processing in the inner segment using the patch clamp technique (suction electrodes can only measure photocurrents generated locally in the membrane enclosed within the pipette).

We have better highlighted our use of two different experimental approaches (in the Abstract and Introduction). Furthermore, we have stated explicitly that inter-photoreceptor signaling can be observed with patch clamp but not with suction electrodes (Results, subsection entitled “The single photon response of SPs is within the range of jawed vertebrate rods”).

*The section on bleaching adaptation is cryptic and not clear. For SP cells, it was described that the sensitivity was higher after regeneration by exogenous chromophore than in control. Considering that the retina was dark-adapted for 24 hours, and that rhodopsin (unlike cone pigment) does not show the property of re-dissociation (thus no chromophore exchange) in darkness, the increase in sensitivity after regeneration must reflect the presence of naked opsin already in darkness due to vitamin-A deficiency as the authors quoted, but they should at least make some clarifying comments and possibly even conduct more experiments to investigate further. Otherwise, the authors would come across as trivializing the phenomenon of bleaching adaptation (whereas it is in fact far from trivial). For LP cells, having done just one experiment is also unacceptable. In any case, what is the bottom line of this section in relation to the overall paper? This reviewer thinks that it should either be deleted completely or more experiments are necessary*.

This section is a crucial premise for the single photons response analysis: if, as the reviewer convenes, migrating lampreys present a vitamin-A deficiency, then to correctly assess the sensitivity of rod-like photoreceptors all visual pigment must be first regenerated. Had we not done this analysis, our paper might have come to the opposite conclusion that lamprey SP photoreceptors do not perform as well as many other vertebrate rods! In line with our focus on SPs (and not LPs) the title of this section in our manuscript was “Bleaching adaptation (i.e. dark adaptation) in SPs”.

We have significantly expanded this section (now titled “SPs display bleaching desensitization”) to better clarify its aims and present additional data, both from our original recordings and from new experiments. Specifically:

a) We have expanded our description of the reasons that brought us to test the effects of exogenous retinal on SPs (Results, “SPs display bleaching desensitization”);

b) For SPs we have included the absolute values, for both control and regenerated cells, of light sensitivity and integration time (same subsection). By performing new experiments we have further increased all sample sizes and have been able to include a comparison of the spectral preference of control and regenerated SPs (in the second paragraph of the aforementioned subsection);

c) Our quantitative data on the effect of pigment regeneration on SP integration time has been supplemented with a new figure panel 3B showing the normalized dim flash responses averaged across cells (see figure and legend);

d) For LPs we have added previously available and new data in support of our test of the effect of 9-cis-retinal in a single LP. These data come from separate groups of LPs, recorded either in control (n=7) or after regeneration (n=7). Their sensitivities are significantly different (p<0.05) (Results, “SPs display bleaching desensitization”);

e) We have given greater emphasis to our discovery that in lamprey SPs, as in jawed vertebrate rods and cones, bleached opsin seems to activate the phototransductive cascade (Discussion).

*In the single-photon-response section, no response-amplitude histogram is shown, despite the many flash trails performed. This should be properly shown and compared with the estimate based on the variance/mean ratio*.

Our exclusive use of the well-established variance analysis method to estimate the single photon response amplitude was explained in the original Methods section, where we stated that: “As in other rod studies in which *a* [the single photon response amplitude] was small relative to total recorded dark noise, we could not perform a reliable analysis of the single photon response with amplitude histograms. However, when both the variance and histogram techniques were employed, similar values of *a* were found (7, 9, 78)”.

Notably, out of the 26 published estimates of rod single photon response amplitudes in jawed vertebrates plotted in our Figure 6, only 6 originated from studies that also included an histogram analysis! Given that lamprey SPs have a signal-to-noise ratio at the lower end of the vertebrate range (our Figure 6) it is not surprising that a histogram analysis was not feasible.

Incidentally, not only did Morshedian and Fain not perform an amplitude histogram analysis, but they also do not show actual records of the variance analysis (as we instead do).

We have included a reference to the above explanatory paragraph at the appropriate location in the Results. Also, we have added a mention to the 6/26 statistic just presented (Methods, “Data analysis and reporting”).

Reviewer #3:

*[…] In summary, the physiological data of the SPs and LPs of the northern hemisphere lamprey obtained by the authors are valuable. However, when these results are interpreted as a part of photoreceptor evolution, the presentation becomes murky. For this reason, I cannot recommend that the paper be published as it stands*.

We thank the reviewer for his/her detailed discussion of the evolutionary interpretation of our findings and for raising the important issue of when lampreys diverged relative to the diversification of visual opsins. We did not discuss this in the original manuscript and agree that it should be included.

Our view is that the most recent literature does not provide support for the postulate on which the reviewer’s criticism is based. In fact, it has been hotly debated up to very recently whether the lamprey line diverged *before, between, or after* the two rounds of whole genome duplication (called ‘2R’) (Kasahara, 2007, [70], [43], Mehta et al., 2013, [76]) that gave rise to the 5 groups of vertebrate visual opsins (46). The possibility that the divergence occurred much after 2R was specifically rejected by Putnam et al. (2008, Nature, p.1069). Most importantly, the authors of the authoritative study reporting the sequencing of the lamprey genome ([76], Nature Genetics, 59 authors from more than 30 institutions) state, regarding lampreys and jawed vertebrates, that the data is “consistent with the divergence of the two lineages *shortly after* the last whole-genome duplication event”. In summary, since the five groups of vertebrate visual pigments arose from 2R (46) and the lamprey line likely diverged shortly after that (see above), it follows that *lampreys are indeed a critical model for the understanding of photoreceptor evolution*.

We have significantly revised the Introduction and Discussion sections to improve our arguments for a critical role of lampreys in our understanding of the evolution of rods and dim light vision. In doing so we have integrated several of the reviewer’s suggestions. In particular, we now explicitly discuss the important issue of timing raised by the reviewer, presenting the most recent literature.